# Prospective Association of Maternal Educational Level with Child’s Physical Activity, Screen Time, and Diet Quality

**DOI:** 10.3390/nu14010160

**Published:** 2021-12-30

**Authors:** Gabriela Cárdenas-Fuentes, Clara Homs, Catalina Ramírez-Contreras, Charlotte Juton, Rafael Casas-Esteve, Maria Grau, Isabel Aguilar-Palacio, Montserrat Fitó, Santiago F. Gomez, Helmut Schröder

**Affiliations:** 1Non-Communicable Disease and Environment Research Group, Barcelona Institute for Global Health (ISGlobal), 08003 Barcelona, Spain; gabriela.cardenas@isglobal.org; 2Health Science Faculty, Blanquerna—Universitat Ramon Llull, 08022 Barcelona, Spain; 3Gasol Foundation, 08830 Sant Boi de Llobregat, Spain; choms@gasolfoundation.org; 4PSITIC Research Group, Psychology, Education and Sport Sciences Department, Blanquerna—Universitat Ramon Llull, 08022 Barcelona, Spain; 5Department of Nutrition, Food Science and Gastronomy, Food Science Torribera Campus, 08921 Barcelona, Spain; catalinaramirez.nut@gmail.com; 6INSA-UB, Nutrition and Food Safety Research Institute, 08007 Barcelona, Spain; 7Endocrinology Department, Institut de Recerca Sant Joan de Déu, 08950 Barcelona, Spain; charlottejuton@gmail.com; 8PhD Program, Food and Nutrition University of Barcelona, 08028 Barcelona, Spain; 9CSM Nou Barris, Programa de Jóvenes, 08042 Barcelona, Spain; rafael.casas@csm9b.com; 10Serra-Hunter Fellow, Department of Medicine, University of Barcelona, 08028 Barcelona, Spain; mgrau@imim.es; 11CIBER Epidemiology and Public Health (CIBERESP), Instituto de Salud Carlos III Madrid, 28029 Madrid, Spain; 12Grupo de Investigación en Servicios Sanitarios de Aragón (GRISSA), IIS Aragón, Preventive Medicine and Public Health Department, University of Zaragoza, 50001 Zaragoza, Spain; iaguilar@unizar.es; 13Cardiovascular Risk and Nutrition Research Group (CARIN), Hospital del Mar Medical Research Institute (IMIM), 08003 Barcelona, Spain; mfito@imim.es; 14CIBER Physiopathology of Obesity and Nutrition (CIBERobn), Instituto de Salud Carlos III Madrid, 28029 Madrid, Spain; 15GREpS, Health Education Research Group, Nursing and Physiotherapy Department, University of Lleida, 25008 Lleida, Spain

**Keywords:** maternal educational level, lifestyle behaviors in children, diet quality, sedentary behaviors, physical activity, prospective cohort study

## Abstract

Evidence has identified unhealthy lifestyle behaviors as the main contributors to obesity in children, so it is essential to identify factors that could influence children’s lifestyles. The objective of the present study was to analyze the association of baseline maternal educational level with child’s physical activity, screen time, and dietary habits at follow-up. This community-based cohort study was carried out between 2012 and 2014 and included 1405 children aged 8 to 10 years old. Maternal educational level was used as an indicator of child’s socioeconomic status. Physical activity, screen time, and dietary habits were assessed by validated questionnaires. The odds of having commercially baked goods for breakfast [OR 1.47 (95% CI 1.03 to 2.10)], going more than once a week to a fast-food restaurant [OR 1.64 (95% CI 1.20 to 2.26)], and taking sweets and candys several times a day [OR 3.23 (95% CI 2.14 to 4.87) were significantly higher among children whose mothers had a lower educational level compared to their peers whose mothers had a higher level. These associations held for taking sweets and candy several times a day after additional adjustment for the corresponding dietary behavior at baseline. Maternal educational level was inversely associated (*p* < 0.001) with child’s screen time at follow up and being in the lowest maternal educational category was associated with an increased odds of surpassing the maximum recommended time of screen time of 120 min per day (OR (95% CI) 1.43 (1.07 to 1.90), *p* = 0.016). Maternal education is a predictor for unhealthy dietary habits and high screen time in children.

## 1. Introduction

In recent decades, the exponential increase in the prevalence of childhood obesity has become a major public health problem [1,2]. From 1975 to 2016, it changed globally from less than 1% to more than 6% and 8% in girls and boys, respectively [3], while in Spain, the prevalence of obesity has reached almost 15% of the child-youth population (as recently reported in [4]). Despite relevant improvements in the prevention of this disease, there is still uncertainty as to the key factors to be addressed.

Evidence indicates that unhealthy lifestyle behaviors, such as low levels of physical activity (PA) [5,6], a low-quality diet [7,8], and increased screen time [9,10], promote the development of obesity [11,12]. Unfortunately, the prevalence of these unhealthy behaviors is especially high in children and adolescents with 80% reporting insufficient level of PA [13]. Moreover, it has been described a drastic increase in the time spent in sedentary behaviors (such as watching television) and a substantial decrease in the time spent in physically demanding activities (such as running) after the age of six [6]. Equally disturbing is the proportion of children that consumes a low-quality diet. According to the WHO European Childhood Obesity Surveillance Initiative (COSI), 67% and 77% of children do not consume fruits and fresh vegetables, respectively, on a daily basis [14]. This discouraging evidence, along with the fact that a low-quality diet tends to endure during later adolescence, demands for a prompt call to action [14,15]. In this line, one of the initial steps is to identify the determinants of detrimental lifestyle behaviors to improve, on a later stage, the design of public health strategies.

Evidence suggests that maternal educational level, measured as a proxy of the child’s socioeconomic status, is related to the child’s diet quality, physical activity, and screen time [16,17,18,19,20,21,22,23,24,25,26,27,28,29]. Cross-sectional data indicates that children whose mothers have a low educational level are at higher risk of reporting an unhealthy diet, a low level of physical activity, and high level of screen time [16,17,18,19,20,21,22]. Cross-sectional data aid in hypothesis generation and establish an association, but prospective studies are needed to determine whether these associations may change over time. However, prospective data is somewhat limited [23,24,25,26,27,28,29]. Therefore, the aim of the present study was to analyze the association of baseline maternal educational level with child’s physical activity, screen time and dietary habits at follow-up, in a Spanish cohort. The hypothesis of this study was that baseline maternal education is predictive for child’s physical activity, and dietary and sedentary behaviors, measured two years after baseline.

## 2. Materials and Methods

### 2.1. Study Population

The current study was a prospective cohort in the framework of the POIBC study (Spanish acronym for Prevention of Childhood Obesity: a community-based model). The complete protocol of the POIBC can be found elsewhere [30]. In brief, the POIBC study was a two-year, parallel trial that assessed the efficacy of the THAO—Child Health Program in the prevention of childhood obesity. This program was developed with the aim to improve child’s lifestyle and to reduce the incidence of overweight and obesity in children. The theoretical framework of the program is based on the attitude–social influence–self-efficacy (ASE) model [31], social marketing strategies used in the public health field [32], and CBI guidelines for obesity prevention [33,34]. The program was implemented by the city council, which appointed a local coordinator. In the intervention cities, the coordinator was selected from the community health department. Up to nine different community activities, such as familiar workshops about eating habits and cooking techniques, were implemented in the intervention cities. The POIBC study included a convenience sample of 2249 children aged 8 to 10 years old from elementary schools (4th and 5th grade) from 4 municipalities (Gavà, Terrassa, Sant Boi de Llobregat, and Molins de Rei) of the autonomous community of Catalonia. The study lasted two academic years (2012–2014) and had an average follow-up of 15 months. After excluding participants with missing data on any of the included variables, a final sample of 1405 children (699 girls and 706 boys) with a mean age of 10.1 ± 0.6 years old were included. Parents written consent was obtained on behalf of all children and the study protocol was approved by the local Ethics Committee (CEIC—Parc de Salut Mar, Barcelona, Spain). The POIBC inclusion criteria were an age range from 8 to 10 years and a signed parent consent.

### 2.2. Meassurement of Exposure

Maternal education level was self-reported and categorized into 3 levels: (i) no schooling or primary school, (ii) secondary school, (iii) technical or higher (graduate-level) university degree.

### 2.3. Measssurement of Outcomes

#### 2.3.1. Dietary Behaviors

Mothers and children responded themselves to the corresponding questionnaires. The diet quality in children was estimated by the KIDMED index [35]. This questionnaire was developed to assess diet quality by considering the principles that support and those that undermine the adherence to the Mediterranean diet. This questionnaire was previously validated [35] in Spanish children in the framework of the EnKid study. It is a 16 items-questionnaire with 4 items indicating non-adherence and punctuated with −1 and 12 items indicating adherence and punctuated with +1. The final score ranges from −4 to 12 points, with higher scores indicating a higher adherence. The items included in the current study were those related with the principles that undermine the MedDiet and included the followings: (i) skips breakfast, (ii) has commercially baked goods or pastries for breakfast, (iii) goes more than once a week to a fast-food restaurant, and (iv) takes sweets and candies several times every day.

#### 2.3.2. Physical Activity

The physical activity questionnaire for children (PAQ-C) was used to assess the levels of PA in the children [36]. It includes nine items, each of the ones scored from one to five points. The final PAQ-C score was the mean of the nine items. Higher scores indicated higher levels of PA.

#### 2.3.3. Screen-Time

Screen-time was assessed by the screen-time sedentary behaviour questionnaire [37], which asks about the time spent in four activities, separately for weekdays and weekends. The activities included (1) watching television, (2) playing computer games, (3) playing console (video) games, and (4) using a mobile phone. Children with equal or more than 120 min of screen time per day were classified as non-compliers of screen time recommendation [38].

### 2.4. Statistical Analysis

Baseline characteristics of the study population were described across the levels of maternal education and general linear models were fitted to obtain *p* values for linear trends. Two statistical models were used to respond to the research question: “is baseline maternal education predictive for child’s behavior at follow-up?” (i) For continuous outcomes, general linear models were applied with baseline maternal education as the fixed factor and screen time and PA at follow-up as the dependent variables. Polynomial contrast was used to estimate *p* for linear trend with a post-hoc Bonferroni correction for multiple comparisons. Models were adjusted for sex, age, zBMI, municipality, school, and allocation to intervention or control group. (ii) For dichotomous outcomes, multiple logistic regression models were used to determine the associations between baseline maternal education and child´s dichotomous outcomes at follow-up including risk of meeting child’s screen time recommendation and presenting an unhealthy dietary habit (skipping breakfast, having commercially baked goods or pastries for breakfast, going more than once a week to a fast-food restaurant, and taking sweets and candies several times every day). Models were adjusted for sex, age, zBMI, municipality, school, and allocation to intervention or control group in the first model and additionally with the corresponding dietary behavior in the second model.

Interactions of sex, age, and allocation to intervention or control group with maternal education were tested. The associations were considered significant if *p* < 0.05. All statistical analysis was performed using SPSS for Windows version 22 (SPSS, Inc., Chicago, IL, USA).

## 3. Results

Table 1 shows descriptive data on the study population according to maternal education. From the 1405 children included in the analysis 26.4%, 39.4%, and 34.2% had mothers with primary education, secondary education, and equal or higher than technical or university education, respectively. Descriptive analysis revealed that children of mothers with higher educational level were somewhat younger, ate healthier, and spent less time in front of a screen compared to their peers whose mothers had a lower level (Table 1). The associations between baseline maternal educational level and children’s level of PA and screen time at follow up are shown in Table 2. Children of mothers with higher educational level showed a significant (*p* < 0.001) lower increase in time in front of a screen compared with children of mothers with lower level. This association was strongly affected by child baseline sedentary time which was significantly higher in the low educated group compared to the high educated group (Table 1).

Figure 1 presents the odds ratio of unhealthy dietary habits of the children in relation with maternal educational level. Children from low educated mothers showed a 1.47 (1.03 to 2.10, (*p* = 0.046), 1.64 (1.20 to 2.26) (*p* = 0.003), and 3.23 (2.14 to 4.87) (*p* < 0.001) higher odds of eating commercially baked goods or pastries for breakfast, going more than once a week to a fast-food restaurant, and taking sweets several times daily, respectively, compared to children from highly educated mothers. These associations held only for taking sweets several times daily after additional adjustment for the corresponding baseline dietary behavior.

Children of mothers with the lowest educational level showed 43% major odds of surpassing the maximum recommended level of screen time of 120 min per day [OR (95% CI) 1.43 (1.07 to 1.90)], *p* = 0.016) at follow-up compared to their peers whose mothers had the highest level.

There was no significant interaction between maternal education, sex, age, and allocation to intervention or control group.

## 4. Discussion

The present study found that a lower level of maternal education was associated with an increased risk of having unhealthy dietary habits in children, including having commercially baked goods for breakfast, going to a fast-food restaurant more than once a week, or taking sweets and candys several times a day. Adjustment for the corresponding dietary behavior at baseline resulted in a loss of significance with the exception of taking sweets and candys several times a day. Furthermore, lower maternal educational level was positively associated with child screen-time and not meeting screen-time recommendations.

Several studies, with both cross-sectional [16,17,18,19] and longitudinal [23,24,25] designs, have shown an inverse association between parental educational level and the time spent in front of a screen in children. These findings are aligned with the results of the present study which additionally showed that a lower maternal educational level was predictive for not meeting screen-time recommendation for children. This is of concern due to the positive relationship of time in front of a screen and obesity in children [10].

The evidence showing the association of maternal education with children’s physical activity level have shown inconsistent results. Cross-sectional data from the HELENA study showed non-significant difference of objectively measured PA between children whose mothers had a lower or higher educational level [39]. In addition, pooled data from 10 studies conducted in Europe, Australia, Brazil, and the USA [40] concluded that adolescents of mothers with lower education may not be at a disadvantage in terms of overall objectively measured PA. This cross-sectional evidence is in concordance with prospective findings from the present study. Conversely, a twin study performed in the Netherlands and Finland [26] evidenced that higher PA levels are more likely to be observed within children whose parents have a higher educational level. Mutz and colleagues found that the socioeconomic status, measured by parent’s mean educational level and income, positively predicts moderate to vigorous PA in children between 8 and 11 years of age [20].

High diet quality during childhood is commonly associated with higher maternal educational level [21,28,41,42], higher socioeconomic status [22,28,43], and less social vulnerabilities [41,42]. Moreover, unhealthy eating patterns, characterized by high intakes of sweet food and drinks, snacks, pastries, or fast food [21,22,28,44] are most commonly observed within children of parents with a lower educational status. Our results are aligned with the existing evidence since children whose mothers had a higher educational level are less likely to carry out unhealthy behaviors such as having commercially baked goods or pastries for breakfast, going more than once a week to a fast-food restaurant, or taking sweets and candies several times every day. Parental practices are crucial to prevent unhealthy food consumption within children [45]. Specifically, a “restrictive guidance” parental practice, characterized by the capacity of parents to set limits, rules, or restrictions regarding food consumption, could be effective in preventing unhealthy eating for children seven years old and older. Mothers with lower educational levels have less opportunities of deploying adequate parenting skills since they could spend less time with their kids [46] and have less parenting support and higher stress levels [47]. In this sense, general family functioning and parental psychological distress is associated with poorer eating habits [48]. Moreover, mothers with a low educational level tend to use more frequently unhealthy foods as an economical accessible reward to their kids in the European context [49]. Using food as a reward from early childhood is prospectively associated with picky eating [49] and with poorer health outcomes [50]. Furthermore, evidence indicates that healthy dietary choices were associated with higher economic costs in children and adolescents [51]. This fact might also partially explain poorer eating habits among children with mothers of low socioeconomic levels.

It has been shown that children whose mothers have a lower educational level are more likely to present a higher screen time, a lower physical activity level, and an unbalanced dietary pattern [52,53,54,55]. It is relevant to highlight that increasing physical activity levels and reducing screen time is longitudinally related to better health-related quality-of-life and socio-emotional outcomes [55] and a reduced risk of non-communicable diseases such as depression [56] or obesity [9,10].

Data on adherence to the Mediterranean diet, PA, and screen time were recorded by questionnaires and therefore prone to the inherent limitations of self-reported data, such as memory bias, misunderstanding, and social desirability. At about eight years of age, children have the cognitive skills to self-report health data [57], and several questionnaires, including those used in the present study, have been designed and validated to collect these data from children aged eight years and older [58]. The mean age of participants in our study is 10 years old and the time invested to answer the questionnaire was usually more than 25 min; the attention span could be shorter at this age than for older children more used to investing time in reading.

Additionally, cognitive skills of children aged 8 to 10 are not fully developed, which could have a stronger effect on response accuracy of self-reported data, compared to adults or older children.

Low maternal education was predictive for child’s poor eating habits, high screen-time, and not meeting screen-time recommendations in Spanish children.

## Figures and Tables

**Figure 1 nutrients-14-00160-f001:**
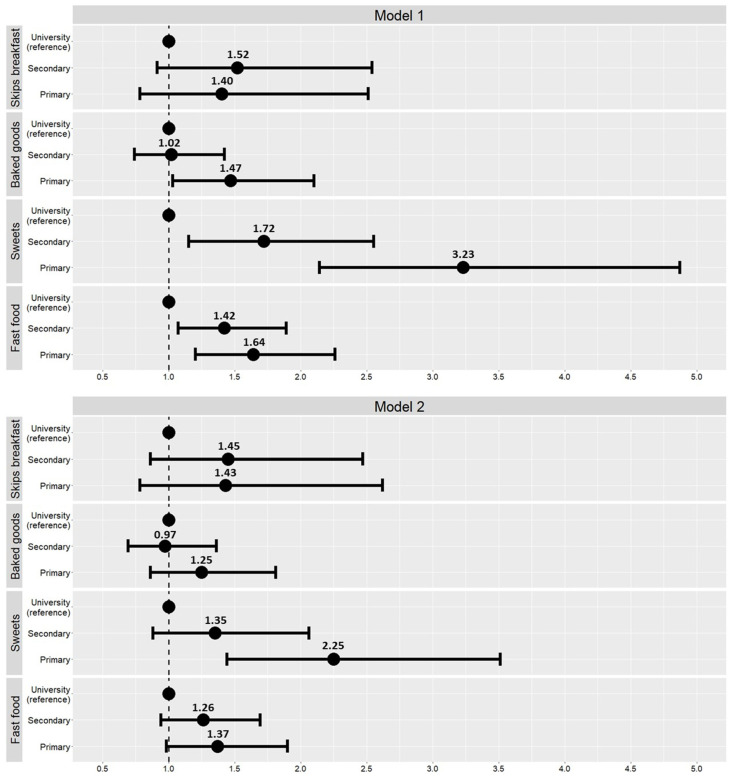
Association of maternal education with children’s having unhealthy dietary habits. Model 1. Multiple logistic regression models adjusted for sex, age, zBMI, municipality, school, and allocation to intervention or control group. Model 2. Multiple logistic regression models adjusted for sex, age, zBMI, municipality, school, allocation to intervention or control group, and the corresponding dietary behavior at baseline. Skips breakfast (no = 0, yes = 1), has commercially baked goods or pastries for breakfast (no = 0, yes = 1), takes sweets and candies several times every day (no = 0, yes = 1), goes more than once a week to a fast-food restaurant (no = 0, yes = 1).

**Table 1 nutrients-14-00160-t001:** Baseline characteristics of the study population across levels of maternal education (*n* = 1405) ^a^.

		Maternal Education		
	Primary *n* = 371	Secondary *n* = 553	University *n* = 481	P for Linear Trend
Sex (%)				
Girls	183 (49.3)	273 (49.4)	243 (50.5)	0.72
Boys	188 (50.7)	280 (50.6)	238 (49.5)	
Age (years)	10.2 (10.1 to 10.2)	10.1 (10.0 to 10.1)	10.1 (10.0 to 10.1)	0.004
zBMI ^b^	0.71 (0.59 to 0.83)	0.73 (0.64 to 0.83)	0.65 (0.54 to 0.76)	0.483
PAQ-C score (unit) ^c^	3.0 (2.9 to 3.0)	2.9 (2.9 to 3.0)	3.0 (3.0 to 3.1)	0.114
Screen time (minutes per day) ^d^	102.9 (49.3 to 200.0)	87.9 (49.3 to 167.9)	81.4 (45.0 to 128.6)	<0.001
Screen time recommendation (%) ^e^	43.3 (38.5 to 48.3)	38.5 (34.5 to 42.5)	28.7(24.3 to 33.0)	<0.001
Skips breakfast (%)	4.0 (1.9 to 6.2)	5.4 (3.7 to 7.2)	4.0 (2.0 to 5.8)	0.949
Has commercially baked goods or pastries for breakfast (%)	28.8 (24.7 to 33.0)	20.3 (16.8 to 23.7)	18.1 (14.4 to 21.8)	<0.001
Goes more than once a week to a fast-food restaurant (%)	25.6 (21.9 to 29.3)	20.1 (16.9 to 23.3)	10.4 (7.0 to 13.8)	<0.001
Takes sweets and candies several times every day (%)	25.6 (21.9 to 29.4)	17.4 (14.7 to 20.4)	8.3 (5.1 to 11.6)	<0.001

^a^ Values are presented as number (proportion), mean (confidence interval), and median (interquartile range) for categorical, continuous normal, and continuous non-normal distributed variables, respectively. ^b^ z-value for BMI. ^c^ The physical activity questionnaire for children (PAQ-C) includes nine items, each one scoring one to five points. The mean of the scores was used as the final PAQ-C score. Higher scores indicate higher levels of PA. ^d^ Screen time included time spent using the computer, watching television, and playing with a gaming console. ^e^ Not meeting recommendations for screen time views (not more than 2 h per day).

**Table 2 nutrients-14-00160-t002:** Association of baseline maternal education with child’s physical activity and screen time at follow-up ^a^.

		Maternal Education		
	Primary	Secondary	University	P Linear Trend
	*n* = 371	*n* = 553	*n* = 481	
PAQ-C score (unit) ^b^	3.0 (2.9 to 3.1)	3.1 (3.0 to 3.2)	3.1 (3.0 to 3.2)	0.324
Screen time (min/d) ^c^	294.9 (180.2 to 209.6)	173.4 (161.7 to 185.2)	147.0 (134.1 to 160.0)	<0.001

^a^ Adjusted for sex, age, baseline zBMI, municipality, school, and allocation to intervention or control group. Values are expressed as means (confidence intervals). ^b^ The physical activity questionnaire for children (PAQ-C) includes nine items, each one scoring one to five points. ^c^ Screen time includes time spent using the computer, watching television, and playing with a gaming console.

## Data Availability

Available on request.

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
