# Peer review of "Prospective Association of Maternal Educational Level with Child’s Physical Activity, Screen Time, and Diet Quality"

_nutrients, 2021, doi:10.3390/nu14010160_

Round 1

Reviewer 1 Report

This is a nice paper, but in my opinion, this is not describing a prospective association, it is describing a cross-sectional association. However, the data of this study could be analyzed in a different way and give us knew knowledge of a prospective association of education on energy balance-related behaviors. The focus should then be on the changes in the behaviors according to maternal educational level, and not on the behaviors at the age of 12. A reason why I’m saying that the analyzed association is cross-sectional is because that maternal educational level will hardly change in two years, so asking the mothers educational level when the children is 10 years old instead of asking when they are 12 years old will not add any value to the results.

I therefor either suggest that the authors will not use the words prospective association in the article at all. Then the title, the introduction, the objective of the study, the methods and discussion should be rewritten according to that. (now, one of the arguments for why conducting this study was the follow up design and the prospective association, and the objective should be changed to only looking at a cross-sectional design). The other suggestion is that the authors should use the valuable data they have in a longitudinal setting and look at the change (15 months) in the behaviors and whether they are different according to the maternal educational level. I this kind of analyses the outcome will be the behavior in 2014, the determinant is the maternal educational level, and the analyses should be adjusted for the baseline behavior in 2012 in addition to adjustments for age, sex and intervention. If the intervention had an effect (there is no mention about that in the article) it would also be interesting to test whether there is an interaction between intervention and maternal educational level, showing that the intervention was more effective in certain educational level group, but that might be out of the scope of this manuscript.   

Other comments are following:

  1. I did not find any mentioned of who did fill in the questionnaires. Was it the child, the mother or someone else? Did the mother report the educational level and the child the behaviors, this was not clear to me?
  2. Why are different dietary variables used for baseline and follow up? Was the whole instrument included in baseline and only the 4 items in follow up? If it so, then I would only use the same for items for baseline and not the total KIDMED INDEX (as showed in Table 1).
  3. The authors have tested for interaction of sex and age with maternal education. I would suggest that they also test for interaction of intervention with maternal education.
  4. Lines 145-147: The Odds ratios and CI should be reported not the percentages.
  5. Table 1 (lines 156-157. Why are baseline results not adjusted for age and sex as for results in follow up, presented in Table 2?
  6. Table 1 and Table 2. I would prefer describing the categories for maternal educational level in same way as in Figure 1 (Primary, Secondary, High). Or then describe they in same way in Figure 1 as in Table 1 and Table 2.
  7. Figure 1. Mentioned also in the Figure which is the reference group.

Author Response

This is a nice paper, but in my opinion, this is not describing a prospective association, it is describing a cross-sectional association.

Response: Thank you.

However, the data of this study could be analyzed in a different way and give us knew knowledge of a prospective association of education on energy balance-related behaviors. The focus should then be on the changes in the behaviors according to maternal educational level, and not on the behaviors at the age of 12. A reason why I’m saying that the analyzed association is cross-sectional is because that maternal educational level will hardly change in two years, so asking the mothers educational level when the children is 10 years old instead of asking when they are 12 years old will not add any value to the results.

Response: We respectfully disagree with the reviewer. This is not a cross-sectional analysis. The exposure in the present analysis was the maternal educational level at baseline and the outcome the lifestyle in the follow-up. In other words, the exposure was measured before the outcome. This is a predictive model and indeed prospective in nature.

I therefor either suggest that the authors will not use the words prospective association in the article at all. Then the title, the introduction, the objective of the study, the methods and discussion should be rewritten according to that. (now, one of the arguments for why conducting this study was the follow up design and the prospective association, and the objective should be changed to only looking at a cross-sectional design). The other suggestion is that the authors should use the valuable data they have in a longitudinal setting and look at the change (15 months) in the behaviors and whether they are different according to the maternal educational level. I this kind of analyses the outcome will be the behavior in 2014, the determinant is the maternal educational level, and the analyses should be adjusted for the baseline behavior in 2012 in addition to adjustments for age, sex and intervention. If the intervention had an effect (there is no mention about that in the article) it would also be interesting to test whether there is an interaction between intervention and maternal educational level, showing that the intervention was more effective in certain educational level group, but that might be out of the scope of this manuscript.   

Response: We agree with the reviewer that there are several possibilities to analyze prospective associations. We decided to choose the predictive model with the exposure at baseline and the outcome at follow-up due to limited statistical power of the model proposed by the reviewer. To analyze changes as proposed by the reviewer we converted each of the dichotomous dietary variables at baseline and follow-up into a dummy variable with 4 categories. For example, Skipping breakfast (SKB):

1) Not SKB-Baseline and Not SKB-Follow-up; n= 1267 (90,2%)

2) SKB-Baseline and SKB-Follow-up. n= 21 (1.5%)

3) Not SKB-Baseline and SKB-Follow-up n=74 (5.3%)

4) SKB-Baseline and Not SKB-Baseline n=43 (3.1%)

But the number of children changing their habits was very small. This fact causes a substantial loss in statistical power. Therefore, we decided to present a predictive model with maternal education as the exposure and dietary behaviors in the follow-up as the outcome. In the revised version of the manuscript, we have included a second model in Figure 1 including the corresponding baseline dietary behaviors and additionally some potential confounders as suggested by reviewer 2. Furthermore, we performed an analysis proposed by the reviewer for our 2 continuous variables namely physical activity and screen time. In this new analysis we included maternal education as the exposure and changes in dietary behaviors as the outcome  (Please see Table 2). Baseline levels of screen-time were substantially different among strata of maternal education. Children of mothers with primary education showed a baseline value of 102 min/d whereas children of mothers with university degree had a baseline value of 143 min/d. The corresponding follow up values were 147 min/d and 194 min/d respectively. We have tested for interaction between baseline values of screen-time and maternal education. The interaction was significant and therefore we decided not to include baseline levels of screen-time as a covariable in the model.

Other comments are following:

  1. I did not find any mentioned of who did fill in the questionnaires. Was it the child, the mother or someone else? Did the mother report the educational level and the child the behaviors, this was not clear to me?

Response: Yes, mother and child have answered these questions themselves. We have clarified this in the text as follows: “Mothers and children responded themselves to the corresponding questionnaires”.  (page: 2   lines: 110-111; clean version of the manuscript).

  1. Why are different dietary variables used for baseline and follow up? Was the whole instrument included in baseline and only the 4 items in follow up? If it so, then I would only use the same for items for baseline and not the total KIDMED INDEX (as showed in Table 1).

Response: For the present analysis we have used these 4 questions of the KIDMED questionnaire. The reviewer is right that the inclusion of the KIDMED index in table 1 is superfluous. We have replaced the KIDMED index by baseline data of the 4 items as proposed by the reviewer

  1. The authors have tested for interaction of sex and age with maternal education. I would suggest that they also test for interaction of intervention with maternal education.

Response: We have additionally tested for interaction with intervention as suggested by the reviewer. We didn´t find a significant interaction with intervention. We have modified the text in the statistical section as follows: “Interactions of sex, age, and allocation to intervention or control group with maternal education were tested”. (page: 3 lines: 145-146)

There was no significant interaction between maternal education, sex age, and allocation to intervention or control group”. (page:4 lines: 172-173).

  1. Lines 145-147: The Odds ratios and CI should be reported not the percentages.

Response: This has been done.

  1. Table 1 (lines 156-157. Why are baseline results not adjusted for age and sex as for results in follow up, presented in Table 2?

Response: Table 1 shows for the present analysis the characteristics of the study population according to maternal education. This is a descriptive data presentation, and hence, not adjusted. We clarified this in the text as follows: “Table 1 shows descriptive data on the study population according to maternal education”. (page: 4 lines: 150-151)

  1. Table 1 and Table 2. I would prefer describing the categories for maternal educational level in same way as in Figure 1 (Primary, Secondary, High). Or then describe they in same way in Figure 1 as in Table 1 and Table 2.

Response: This has been done.

  1. Figure 1. Mentioned also in the Figure which is the reference group.

Response: This has been done.

Reviewer 2 Report

The manuscript submitted for publication to Nutrients by Cárdenas-Fuentestitled: “Prospective association of maternal educational level with child’s physical activity, screen time, and diet quality ” is a study aimed todiscuss investigate the association of mothers’ educational level and children’s physical activity, screen time and diet quality in Spanish children.

This is an interesting topic with important public health implications. The manuscript is well-written with good flow and language, well organized and easy to follow for the reader.

The reviewer would like to offer the following points for improvement:

  1. Consider elaborating the introduction. Include statistics pertinent to the world and Spain as per trends seen relevant to the topic.
  2. Consider articulating a hypothesis at the end of the introduction section.
  3. The study seems to be drawing data from a larger study performed in Catalonia. Could the authors provide more information on the region and its population. Is there a specific reason this region is selected and/or does its population has specific characteristics of interest in regards to the research questions?
  4. Could the authors mention inclusion and exclusion criteria?
  5. What were potential confounding factors? Were they addressed?

Author Response

This is an interesting topic with important public health implications. The manuscript is well-written with good flow and language, well organized and easy to follow for the reader.

Response: Thank you for this assessment of our work.

The reviewer would like to offer the following points for improvement:

  1. Consider elaborating the introduction. Include statistics pertinent to the world and Spain as per trends seen relevant to the topic.

Response: We have modified the introduction according to the suggestion of the reviewer. Four new references have been included. (page: 2  lines: 60-91; in the clean version of the manuscript)

  1. Consider articulating a hypothesis at the end of the introduction section.

Response: We have added the hypothesis at the end of the introduction as suggested by the reviewer. “The hypothesis of this study was that maternal education is predictive for child’s physical activity, and dietary and sedentary behaviors”. (page: 2  lines:90-91)

  1. The study seems to be drawing data from a larger study performed in Catalonia. Could the authors provide more information on the region and its population. Is there a specific reason this region is selected and/or does its population has specific characteristics of interest in regards to the research questions?

Response: We have stated in the Method section that “The current study was a prospective cohort in the framework of the POIBC study (Spanish acronym for Prevention of Childhood Obesity: a community-based model). The complete protocol of the POIBC can be found elsewhere [27]. In brief, the POIBC study was a two-year, parallel trial that assessed the efficacy of the THAO – Child Health Program in the prevention of childhood obesity.”  

The POIBC study was an intervention study addressing the effect of a childhood obesity prevention program. Hence, there was no specific reason why we selected this population regarding the current research question. It is quite common that data from large intervention studies are used for prospective analysis to address relevant associations (for example PREDIMED and PREDIMED Plus intervention trials). Recruitment of the children and their parents was performed in schools of 4 municipalities (Gavà, Terrassa, Sant Boi de Llobregat, and Molins de Rei) from Catalonia (Spain). We added this information to the method section: “The POIBC study included a convenience sample of 2,249 children aged 8 to 10 years old from elementary schools (4th and 5th grade) from 4 municipalities (Gavà, Terrassa, Sant Boi de Llobregat, and Molins de Rei) of the autonomous community of Catalonia”.  (page:3; lines 97-100).

  1. Could the authors mention inclusion and exclusion criteria?

Response: We have added the following text to the method section: The POIBC inclusion criteria were an age range from 8 to 10 years and a signed parent consent. (page: 3 ; lines 105-106).

  1. What were potential confounding factors? Were they addressed?

Response: We have adjusted all analysis, with the exception of descriptive data of the study population, for age, sex, municipality, allocation to intervention or control group, and child’s baseline zBMI.

Round 2

Reviewer 1 Report

I can see that the authors have considered some of my comments and suggestions, but they did not agree on what I think is the largest problem in the article, that this is mainly a cross-sectional study not a prospective study. The authors think that that when the education is measured at baseline and the behavioural outcomes on follow up this is the same as a prospective study. I still think they are mainly doing a cross-sectional study (only exception is Table 2 in the new version). There is very small possibility that the parent’s educational level will change between the baseline and the follow up. You could also argue that you are doing a prospective study if you measure the gender of the child at baseline and then the behaviours at follow up, but as for educational level I would argue that that would be a cross-sectional study, because gender will not change between baseline and follow up, only if you look at a change in the behaviour you could call it a prospective study. But of course, you could argue that all studies assessing the associations between gender or parental educational level with children’s behaviours are prospective because gender is defined at birth and parental educational level is not changing much in time, but then it is indifferent if the determinants are measured at baseline or at follow up.

I also noticed that the authors have included new confounders, children’s weight and children’s behaviours, in the analyses, suggested by reviewer 2. The main reason for adjusting for confounders is that a confounder influences both the dependent variable and independent variable, causing a spurious association. In this study the independent variable is parental educational level and the dependent is the child’s behaviour. I don’t understand why the authors think that children's weight and children's other behaviours are confounder and therfore adjusted for children’s weight and children’s behaviours.  In which way could children’s weight influence on parental educational level?  And in which way could children’s other behaviours influence on parental educational level? I suggest that the analyses are not adjusted for these variables, because I think it can cause overadjustment bias and unnecessary adjustment. The authors also mentioned that there are problems with statistical power in the models, another reason for not conducting unnecessary adjustment.

In the authors answers they reported following “We have tested for interaction between baseline values of screen-time and maternal education. The interaction was significant and therefore we decided not to include baseline levels of screen-time as a covariable in the model.” I found this result very interesting and I think this could be one of the main results in the study. I don’t understand the argument that he authors therefore did not include baseline levels of screen-time as a covariable in the model. The result showed that the change in screentime from baseline to follow up was different in different parental educational groups. This kind of analyses I think is showing that depending of which parental educational level children have when they are 10 years old will influence on how the screentime will change until they are 12 years old dependin on parental educational level. I would be very interesting to see in which parental educational level group is the change larger and in which is it smaller. This can be shown as an figure.

If the authors still want to claim that this study is prospective, then they should highlight that this is because of the analyses showed in table 2 and I also suggest that they include the analyses they have conducted where they have tested for interaction when they have analysed the change in screentime (dependent variable the follow up screen time, independent parental educational level, adjusted for baseline screen time and tested for interaction baseline screentime*parental educational level).   

Reviewer 2 Report

Authors made reasonable efforts in addressing reviewer’s points.

Author Response

Thank you